# Collaboration for Developing and Sustaining Community Dementia-Friendly Initiatives: A Realist Evaluation

**DOI:** 10.3390/ijerph20054006

**Published:** 2023-02-23

**Authors:** Marjolein Thijssen, Maud J. L. Graff, Monique A. S. Lexis, Maria W. G. Nijhuis-van der Sanden, Kate Radford, Pip A. Logan, Ramon Daniels, Wietske Kuijer-Siebelink

**Affiliations:** 1Radboud University Medical Center, Radboudumc Research Institute, Scientific Center for Quality of Healthcare (IQ Healthcare), 6525 GA Nijmegen, The Netherlands; 2Radboud Alzheimer Center, Radboud University Medical Center, 6525 GA Nijmegen, The Netherlands; 3Department Occupational Therapy, School of Allied Health, HAN University of Applied Sciences, 6525 EN Nijmegen, The Netherlands; 4Research Centre Assistive Technology in Care, Zuyd University of Applied Sciences, 6419 DJ Heerlen, The Netherlands; 5Centre for Rehabilitation and Ageing Research, School of Medicine, University of Nottingham, Nottingham NG7 2TU, UK; 6Nottingham City Care Partnership, Nottingham NG6 8WR, UK; 7School of Education, HAN University of Applied Sciences, 6525 EN Nijmegen, The Netherlands; 8Radboudumc Health Academy, Research on Learning and Education, Radboud University MEDICAL Center, 6525 GA Nijmegen, The Netherlands

**Keywords:** collaboration, dementia-friendly, inclusion, realist evaluation, patient and public involvement, inequity, social participation

## Abstract

Background: Dementia-friendly communities (DFCs) are seen as key to the inclusion and participation of people with dementia and carers. Dementia-friendly initiatives (DFIs) are important building blocks for the growth of DFCs. The collaboration between different stakeholders is a central aspect in developing and sustaining DFIs. Aim: This study tests and refines an initial theory about collaborating for DFIs with special attention for the involvement of people with dementia and their carers during the collaboration for DFIs. The realist approach is used for deepening contextual aspects, mechanisms, outcomes, and its explanatory power. Methods: A participatory case study design using qualitative data (focus groups, observations, reflections, minutes from meetings, and exit interviews) was executed in four Dutch municipalities that have ambitions to become dementia- friendly communities. Results: The refined theory on the collaboration for DFIs incorporates contextual aspects such as diversity, shared insights, and clarity. It draws attention to the importance of mechanisms such as the recognition of efforts and progress, informal distributed leadership, interdependency, belonging, significance, and commitment. These mechanisms resonate with feeling useful and feeling collectively powerful in the collaboration. The outcomes of collaboration were activation, getting new ideas, and fun. Our findings address how stakeholders’ routines and perspectives impact the involvement of people with dementia and their carers during collaboration. Conclusion: This study provides detailed information about collaboration for DFIs. The collaboration for DFIs is largely influenced by feeling useful and collectively powerful. Further research is needed to understand how these mechanisms can be triggered with the involvement of people with dementia and their carers in the heart of the collaboration.

## 1. Introduction

Health disparity refers to adverse health differences affecting marginalized groups arising from systemic factors that lead to social disadvantage [1,2]. Health disparities are an equity issue and reflect both gaps in the quality of care received and broader patterns of injustice within society [2]. They affect people who have systematically experienced social or economic obstacles to health based on, for example, their ethnic group, socioeconomic status, gender, age, mental health, or a cognitive, sensory, or physical disability [2]. People with dementia are increasingly recognized as a health disparity population [3]. There were over 50-million people worldwide living with dementia in 2020. This number will almost double every 20 years, reaching 82 million in 2030 and 152 million in 2050 [4]. Dementia affects social skills, behaviour, the functioning and activities of daily living, and the ability to relate to others [5,6]. Social relationships have a significant impact on the quality of life of people with dementia [5,7]. Research aimed at maximizing the health and inclusion of people with disabilities, such as dementia, is crucial [8]. The resources needed for health and inclusion require a high quality of medical care, physical and social conditions in homes, neighbourhoods, and workplaces, and of education of stakeholders [1,2,9,10].

Dementia-friendly communities (DFC) are a growing response to the above mentioned need [11]. Dementia-friendly communities (DFCs) are locations (for example, a city or neighbourhood) or organizations with a specific focus (for example a workplace [12,13]) that aim to ensure that people with dementia and their carers are included and valued as equal citizens [10,14,15]. Additionally, it means that people with dementia are entitled to equal opportunities in all aspects of life and equal access to public services and space [16]. DFCs represent an ongoing process of learning and culture change, rather than a singular “state” [6]. The advancement of DFCs has a similar approach internationally [6] and requires both top-down input by the (local) government, such as policy, facilitation, and finances, and bottom-up (local) resources and dementia-friendly initiatives (DFIs), such as initiatives focusing on awareness about dementia and related social interaction [10,13,17,18]. Such initiatives are the “building blocks” in the advancement of DFCs [13,19]. DFIs are initiatives or activities that aim to promote dignity, empowerment, engagement, and autonomy to enhance the well-being of people with dementia and their carers, and to address the needs of carers throughout the dementia trajectory [19]. Examples of DFIs exist in three categories: (1) Dementia-specific initiatives, such as education in dementia or adapting the social and physical environment for people with dementia; (2) Dementia-inclusive initiatives, such as physical exercise or practicing creative activities in an existing group; and (3) DFIs in which people with dementia are both co-organizers and executers, for example, music events or theatre [13,20].

The development of DFIs is a complex process. It requires the commitment of key actors in a local public context, ranging from municipalities, healthcare, and social work organizations to businesses and voluntary-sector organizations [6,10,21,22,23], as well as the involvement of people with dementia and their carers [6,16]. This complexity, and the multiple actors, makes collaboration a key principle for the development and sustainment of community DFIs [16,20,21,24]. Collaboration in a local community context has been studied widely. Multiple studies with different methodologies have provided insight into facilitators and barriers for collaboration by reflecting on the collaboration used for DFIs [6,13,16,21], DFCs [6,10,17,18], age-friendly communities [25,26], and health [27]. In these studies, facilitators, such as including a diversity of partners for different perspectives [6,10,13,16,17,18,21,25,26,27], including strengths and resources [6,10,13,18,25], and barriers, such as having limited long-term (financial) resources [6,17,21,25,27] and a competing commitment of staff [17,21,25,27], are found to be of importance for the success of collaboration in the local public context. Furthermore, the importance of including the target groups as communicators or leaders is stressed. However, the centrality of involvement was not clearly articulated [6,10,16,17,21]. All these studies acknowledge the importance, plurality, and complexity of collaboration in the local context, and the need to understand the influence of the context on the success of the collaboration.

Although the importance of insight into the collaboration and facilitating and hindering factors is often mentioned, the literature does not explain how collaboration for DFIs works, i.e., which mechanisms are important and which outcomes represent a successful collaboration for DFIs. In our previous study of the Mentality project, we made a first attempt at explaining such causal relationships in an initial theory of collaboration during the development and sustainment of DFIs, based on an analysis of four Dutch best practices of DFCs [28]. Our initial theory about collaboration highlights how contextual aspects such as a personal connection with dementia, the diversity among stakeholders, and the transparency about budget and manpower, as well as mechanisms such as a personalized role, sharing information and joint decision-making lead to feelings of control, connection and being part of a network, evolving into outcomes of collaboration such as taking initiative and/or continuing commitment to a DFIs, satisfaction, purposefulness, and enjoyment in collaboration [28]. Insights on the involvement of people with dementia and their carers were limited in the initial theory of collaboration [28].

Regarding the importance and plurality of the local context in the collaboration for DFIs, more in-depth knowledge of the influence of local contexts and subsequent mechanisms and outcomes on collaboration for DFIs is needed. Additionally, more knowledge is needed about the involvement of people with dementia and their carers in collaboration for DFIs. This study builds on our initial theory about collaboration to deepen the key contextual conditions, mechanisms, and outcomes, as well as their interactions with a special attention on the involvement of people with dementia and their carers. Such insights will improve the understanding of how collaboration works and can best be leveraged to support the inclusion of people with dementia and their carers in the community to reduce health inequalities of a health disparity population. Therefore, the aim of this study is to test the initial theory about the collaboration for DFIs, with special attention for the role of local context and subsequent mechanisms and outcomes, and for the involvement of people with dementia and their carers during this collaboration.

One of the methodological approaches that is suited to examine the contextual influences on collaboration for DFIs is the realist approach. The realist approach facilitates the deeper understanding of what works, for whom, how, and in which context [29,30,31]. It involves the search for causal relations between contexts, underlying mechanisms, and their outcomes [30]. Context refers to pre-existing contextual structures and networks; mechanisms pertain to forces and powers that lead to change; and outcomes are (un)intended results of an intervention. See Box 1 for more detailed definitions of context, mechanisms, and outcomes.

Box 1With detailed realist definitions: Context—Mechanisms—Outcomes.**Context** refers to the backdrop of an intervention [32]. Context includes the pre-existing organisational structures, including the nature and scope of pre-existing (in)formal networks; the cultural norms and history of the community, such as traditions and habits; and former relevant experiences, such as experience with dementia-friendly initiatives [32].**Mechanisms** are not interventions. They are the—often invisible—forces, powers, processes, or interactions that lead to (or inhibit) change. They can be found in the choices, reasoning, and decisions that people make as a result of the resources; the interactions between individuals or groups; and the powers and liabilities that things, people, or institutions have as a result of their position in a group or society [33]. Mechanisms are “triggered” when (program) resources (e.g., information about collaboration, expertise in dementia) interact with specific features of the context (individual, interpersonal, organizational, or institutional) [29].*Mechanism resources* refer to what is triggered in the context of participants/stakeholders [32,34] e.g., knowing how to communicate with people with dementia and sharing information about this.*Mechanism responses* refer to the responses of the participants, all that suggests a change in people’s minds and actions [32,34] e.g., feeling confident in initiating contact and communication.**Outcomes** are either intended or unexpected intervention outcomes, i.e., the result of how people react to the mechanisms, e.g., taking initiative during collaboration. Outcomes can be proximal, intermediate, or final [35,36].

## 2. Methods

Realist Evaluation Cycle

We followed the four phases of the realist evaluation cycle, as illustrated in Figure 1 [30,37].

The application and associated steps of each phase [38,39] are described in the next paragraphs.

## 3. Phase 1: Formulation/Testing of Initial Middle-Range Program Theories

A program theory describes, in words or diagrams, what is supposed to be done in a policy or program (theory of action) and how and why that is expected to work (theory of change) [40]. In our previous study, we formulated three initial middle-range program theories (MRPT) about development and sustainment of DFIs based on the analysis of four Dutch best practice cases [28]. Formulating realist programme theories at the midrange level, such as MRPT, enabled both the specification of contexts, resources, responses leading to outcomes, and the conceptualization and explanation of those outcomes [41]. In this study, we focus on the initial MRPT concerning collaboration with a focus on the interpersonal level during the development and sustainment of DFIs [28]. The initial middle-range program theory, which we found in the aforementioned study, is presented by a figure and corresponding description in Box 2.

Box 2Initial middle-range program theory (MRPT) about collaboration.Initial middle-range program theory: Collaboration 

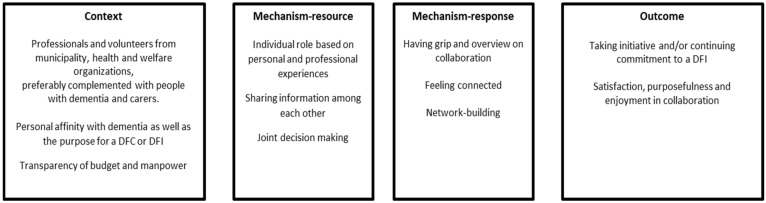

 For collaboration in developing and sustaining DFIs, professionals and volunteers from municipalities and health and welfare organisations came together, preferably complemented by people with dementia and their carers. They all shared a personal connection with dementia or the purpose of a DFC or DFI from various experiences. Other contextual features are transparency about manpower and available budget. As a result, the professionals and volunteers involved knew the conditions to come together. Follow-up actions involved organizing regular meetings about the DFIs to be developed or sustained. At these meetings, professionals and volunteers took roles that best suited their personal and professional experiences. During the meetings, relevant information for building DFIs was shared. Accordingly, meetings were characterised by sharing and, subsequently, joint decision-making. These contextual features and resources on the interpersonal level led to responses of both having control and an overview of collaboration, and of feeling connected with each other. Such responses led to mutual network building and, therefore, brought changes on the interpersonal level. Intermediate outcomes of collaboration were taking initiative and/or continuing commitment to a DFI and experiencing purposefulness, as well as satisfaction and fun during the collaboration itself.

## 4. Phase 2: Choice of Study Design, Data Collection Methods and Procedure

### 4.1. Study Design

Within this realist evaluation, we integrated a participatory action design with a case study design. A case study design allows researchers to explore a “phenomenon within its real-life context” [42]. As such, it allows a holistic in-depth investigation whereby different sources of information and data collection methods can be used concurrently [42]. Next, a participatory action design was appropriate for generating and sharing knowledge by stakeholders’ involvement and engagement [32,43]. For this, the research team consulted the advisory panel of Mentality.

### 4.2. Selection of Cases

To enhance the rigor and robustness of testing, both the advisory panel and research team agreed that, as cases, at least four different municipalities with an ambition for becoming a dementia-friendly community and, thus, developing and sustaining DFIs would need to be analyzed. Furthermore, they agreed that geographic dispersion and the influence of rurality were important for their influence on contextual aspects. We purposively identified two rural sites and two at urban sites in different Dutch regions that were aiming to become dementia friendly. Analyzing more than four case studies was not possible within the timeframe of this study.

After this initial selection, information about the study and its focus on collaboration was sent to the policy officer who was involved in the development of dementia-friendliness in their case. The information enclosed a criterium for inclusion; a commitment to implement the initial MRPT about collaboration during the study period was a prerequisite to be able to test this theory. Policy officers consulted stakeholders in their network, such as social and health professionals who were concerned in dementia and community work, before agreeing to participate. After this, all cases agreed on the importance of collaboration and the participation in this study. A letter of agreement was signed by the local policy officer from each case.

### 4.3. Implementing the Initial MRPT

The implementation of the MRPT involved three resources. The first resource was the set-up of a community of practice (CoP) in each case to develop DFIs. A CoP is a group of people with a collective concern or passion who share and deepen their expertise and tacit knowledge through sharing insights and an ongoing, joint decision-making process [44,45]. By setting up a CoP, contextual features and resources of the initial MRPT, such as organizing regular meetings characterized by sharing and joint decision-making, were implemented. With the policy officer’s approval, the researchers (MT and ML) used a snowball approach to assemble relevant stakeholders in each case. They recruited others from their network to create a group of people who wanted to collaborate for the development and sustainment of DFIs in their community. Relevant stakeholders were professionals and volunteers from healthcare and social care, entrepreneurs, policy officers, and carers of people with dementia.

The CoPs met on a regular basis; on average, 10 times a year. Meetings were online from March 2021 to August 2021 due to the lockdown during the COVID pandemic; otherwise CoP members met face-to-face. Meetings followed a cyclical structure with iterative phases in which plans for DFIs were developed and adjusted if necessary.

The second resource was the appointment of a facilitator for each case. Facilitators were appointed by the research team and had no previous involvement in the case. Facilitators prepared and led meetings, monitored the process by taking notes, and pursued learning and sharing insights by stimulating people to provide input. Facilitators aimed to make themselves redundant by the end of the study. Therefore, they took on tasks together with CoP members who were interested and let them take over and perform tasks, such as leading and preparing meetings, stimulating sharing observations, and/or pursuing reflections and learning during the collaboration. As such, CoP members were involved in these tasks from the start of this study and increasingly took on more responsibility. By appointing a facilitator, resources from the initial MRPT on collaboration were implemented, such as sharing information for building DFIs, sharing insights, joint decision-making, and taking roles that best suited personal and professional experiences. Facilitators stimulated CoP members to invite people with dementia and their carers to join the collaboration. They used reflective questions, such as “How do we know if this is what the people with dementia or their carers really need or want?”; “How can we involve people with dementia and their carers to make them feel included already and not only at the DFI?”; or “What would you like to know/need to know from people with dementia and their carers when developing the DFI?” These were followed by questions or suggestions to come to action, such as, “How are we going to achieve this?” or “What do you need to do this?” By asking reflective questions and suggestions, we wanted to study the local approach regarding the involvement of people with dementia and carers into the collaboration.

The third resource was a reflection tool consisting of a logic model of the MRPT of collaboration [28]. The reflection tool was used by the facilitator to reflect with CoP members on the ongoing collaboration. Hereby, resources of the MRPT, such as sharing relevant information for building DFIs, were implemented.

### 4.4. Range of Collaboration

Within each case, different ways of collaboration occurred to develop or sustain DFIs. First, plans were made for DFIs in the CoP. The input for these plans came from various sources, such as experiences of the CoP members with people with dementia and their carers, former (un)successful collaborations in the community, current local policy plans, and existing information about the experiences and wishes of people with dementia and their carers and other community members regarding activities in their community. Their input was also collected during visits, spontaneous encounters, and activities in the community such as Alzheimer cafes or leisure activities. Each case set its own priorities and aims based on their own information from the community. Cases decided to develop new DFIs and/or to resume former initiatives and activities, e.g., by making a former activity such as a music group for the elderly more dementia-friendly.

Second, collaboration also took place in so-called working groups in which professionals, volunteers, and carers collaborated for the implementation and enactment of the DFI. In these working groups, members of the CoP with an affinity with a specific DFI participated together with other professionals, volunteers, entrepreneurs, and carers, such as volunteers who assisted during a dementia-friendly walking project. Working groups met each other outside CoP meetings, depending on the DFI and required preparation and organization.

Lastly, collaboration took place outside meetings. Such collaboration was often initiated by asking for information, support, or advice that would enable the implementation and enactment of DFIs, e.g., ask an entrepreneur for advice regarding a location when setting up a training dementia-friendliness in the community or ask people with dementia and carers to share their stories during a DFI.

During the study, some collaboration partners joined, and some withdrew from the collaboration. Table 1 provides an overview of all collaboration partners in the range of collaboration.

Within this range of collaboration, communication took place between the members of the CoP and working group and was shared with other collaboration partners when necessary or appropriate. The facilitator was present during meetings of the CoP and the working groups. The researcher was a non-participatory observer during the CoP meeting. Minutes of both CoP and working group meetings were kept in a log and included the following subjects: participants of meetings, undertaken and intended actions, decisions made, and follow-up. The information about contacts with collaboration partners during the implementation of DFIs was shared during meetings and was also kept in the same log. Lastly, information about the involvement of people with dementia and their carers was also kept in the log, e.g., when people with dementia and/or carers visited meetings or when they were involved in implementing the DFI.

The researcher and facilitator also observed and reflected during all collaboration types. These reflections and observations were noted by both the facilitator and the researcher. They contained actual or former events that might influence collaboration, such as experiences during former collaborations or current experiences and preferences about how to communicate and with whom. The information from the logs and minutes was used by the facilitator to support the reflection on collaboration. Next, the information was input for data collection.

### 4.5. Data Collection

Both the advisory panel and the research team agreed that theory-driven focus groups with stakeholders from the cases should be the main source for data collection for testing and refinement of the MRPT. Focus group interview questions were derived from the MRPT and included probing into participating observations, reflections, and the minutes of meetings, in addition to the involvement of people with dementia and their carers during the collaboration for DFIs. See Appendix A for an outline of the focus group interview format.

In each case, the testing of the initial MRPT was introduced in a CoP meeting in January 2021 and took place consecutively over a 12-month period. Initially, Case 1 was included, but after the start of the study, it turned out that they could not comply with the inclusion criteria regarding setting up a CoP and the urgency to develop new DFIs. Therefore, collaboration was ended by mutual agreement; we excluded Case 1 and decided on individual exit interviews in May 2021. For the other three cases, data collection took place between January 2021 and January 2022. The data collection process is described in more detail below.

### 4.6. Data Collection Case 1

Individual exit interviews were held by members of the project team with a policy officer (LD) and the project leader of the DFI Alzheimer café (ML), who were involved in efforts to develop a CoP. Topics during the exit interviews were actions, expectations, and motivation in collaboration and in developing a CoP, the current and past contextual factors that influenced the collaboration and urgency to develop DFIs. See Appendix A for an outline of the exit interview content. Interviews were recorded on Teams and transcribed. They lasted between 50 and 70 min.

### 4.7. Data Collection: Cases 2–4

Focus groups in three other cases were held in two rounds, in July 2021 (T1) and December 2021 (T2), and organized by the researchers (MT Case 1–2 & ML Case 3). Since our study took place during the COVID pandemic, focus groups on T2 were held online, while other focus groups were on location where CoP meetings were held.

Participants were collaboration partners who had been active in the development, execution, and/or sustainment of DFIs. They were selected for each focus group based on their background, role, and input in developing DFIs in order to have a diversity of backgrounds, experiences, and perspectives. They were personally invited during CoP and working group meetings and/or by email. See Table 2 for the presentation of participants per case.

Prior to each focus group interview, the researcher created a visual timeline of activities that were undertaken by collaboration partners based on the minutes, reflections, and observations on the range of collaboration during the previous period. All of this was used as input during the focus group interview. For example, the interviewee asked about differences between the discourse (what was said during the focus group) and the actual practices (what collaboration partners had (not) done, according to the minutes, reflections, and observations) [46]. Manzano’s teacher–learner approach was used to realize an open style of interviewing in the initial stages to generate participant-led insights before moving to more theory-driven questions [39,47]. After the first round, data were analyzed and shared in the next CoP and working group meeting as a member check. Participants also shared insights and, if necessary, adapted plans. Focus group discussions lasted between 90 and 120 min and were audio-recorded, or were recorded on Teams, and transcribed.

See Table 2 for the presentation of the DFIs that were developed or sustained in each case, and the participants during data collection.

The involvement of people with dementia and their carers was mapped, using the typology of Arnstein [48], based on the minutes, reflections, observations, and focus groups.

## 5. Phase 3: Data Extraction and Synthesis

Data extraction and synthesis from the focus groups were conducted in three steps: data preparation, Context–Mechanisms–Outcome configurations (CMOc) extraction and elicitation, and data synthesis.

### 5.1. Data Preparation

Transcripts from the focus group and exit interviews were read by MT (CS 1–4) and ML (CS 1 and 4), and annotations were made to note initial observations relating to the initial MRPT.

### 5.2. CMOC Extraction and Elicitation

Data were extracted from each focus group within each case using Context–Mechanisms–Outcomes configurations (CMOCs) in a data extraction form in Microsoft Excel [30]. The same definitions of contexts, mechanisms, and outcomes, as used in previous phases of this study, were adopted to ensure consistency and transparency in the development of the initial MRPT [34,35,49,50]. See, for definitions, Box 1. Using deductive reasoning [51,52], data extraction was based on the initial MRPT from which the descriptions of the context, mechanisms, and outcomes were drawn in the previous study [46]. If relevant data did not fit into the descriptions based on the MRPT, they were extracted into a new CMOC using inductive reasoning [51,52]. Each CMOC was supported by quotes from the focus groups or exit interviews. We contrasted confirmatory data with rival theory statements from our data (i.e., how the same resources can trigger different responses and outcomes) [53]. Such rival statements could be very informative about underlying mechanisms. An example of a CMOC with confirmatory and rival statements is given in Box 3.

Two researchers (MT and ML) separately extracted CMOCs in Cases 1 and 4 and discussed their findings to ensure consistency and alignment.

Box 3Example of a CMOC with both confirmatory and rival statements.Context–Mechanisms–Outcomes configurations:When the municipality uses flexible regulations for developing and executing DFIs (C), people receive tailored facilitation and information (M), which creates responses such as feelings of trust and support (M), leading to increased motivation (O).Confirmatory statements: *They* [municipality] *are about money, or they know better what the policy is in a municipality or something. I have no idea at all. So, I always find that a welcome addition. It’s nice, yes, just that [it] gives a little bit [of] extra support that motivates me.* (N-1)*… Somewhat looser, especially from the municipality, not clinging to all kinds of rules and things like that, [which] ultimately do not benefit us or the citizens. So that you often get in the way of a lot of spontaneity. That would also be an improvement.* (M-T1)
Rival statement 
*I wish they* [municipality] *were clearer regarding how things can be financed and arranged. I hear constantly: Let’s find out how it can be arranged without getting a straight answer. I find it very confusing and irritating.* (W-T2)

### 5.3. Data Synthesis

The data synthesis included iterative phases by the primary researcher (MT) that were supported by the researcher (ML) and the other members of the research team. Data synthesis proceeded within each case by clustering similar outcomes [54]. Second, commonalities of mechanisms and contexts were also clustered. For example, mechanisms referring to feelings of being taken seriously, as well as contextual aspects such as involvement and connection with different organizations in the community. Based on these clusters and patterns in outcomes, mechanisms and contexts were outlined. An example of such a pattern was having a personal connection or experience with dementia (contextual factors), which led to an understanding of the need for DFCs (mechanisms) and motivation to become active for a DFI (outcomes). These patterns were compared with corresponding configurations and quotes to check for consistency and explanatory power [38]. For the consistency of data synthesis within cases, two researchers (MT and ML) separately synthesized CMO configurations in Cases 1 and 4 and discussed their findings for alignment. Following the aligned configurations, synthesis for Cases 2 and 3 was done by MT.

After within-case synthesis, cross-case synthesis determined if and how the same or different mechanisms occurred in different contexts, leading to insight into which outcomes were built on patterns across cases [42]. This process was supported by an interactive meeting during which the research team and advisory panel reflected on initial patterns and discussed theoretical assumptions [42,46,52]. This reflection and discussion of theoretical assumptions deepened the synthesis, especially the mechanisms and outcomes. For example, a within-case outcome motivation was unpacked by acknowledging different mechanisms, such as belonging and significance, and outcomes, such as activation. For the reflection and discussion, abductive reasoning was used, which involves an iterative process of examining evidence and developing hunches or ideas about the causal factors linked to that evidence [55].

## 6. Phase 4: Theory Testing

The realist evaluation loop was closed by testing and comparing findings to the initial MRPT. This consisted of two steps: retroductive reasoning and refinement, and verification of a refined MRPT, which are explained below.

### 6.1. Testing and Comparing Findings to the Initial MRPT

Using the prior reflection and discussion of Step 3.3, the initial patterns were reviewed by the primary researcher (MT) to determine how they aligned with the initial MRPT. For this, retroductive reasoning was used for the identification of hidden causal forces that lie behind identified patterns or changes in those patterns [52,56]. Retroduction uses both inductive and deductive reasoning and involved moving back and forth across each case while checking the patterns of regularity with the initial MRPT. This resulted in eight patterns. See Appendix A for eight pattern cross cases. Next, the primary researcher (MT) created a presentation for the research team to present each of the eight patterns, which was supported by illustrative quotes and original cases. Interdependencies between other patterns, such as links and ripple effects, were annotated and discussed to support the elucidation of patterns of generative causation until there was a consensus. Ripple effects occur when the outcomes of one configuration become (an aspect of) the context in another configuration [32]. According to Gilmore’s guidelines [38], when a decision was made as to whether data supported, refuted, or refined the initial MRPT, the primary researcher made a note of why this decision was made.

### 6.2. Refinement and Verification of a Refined MRPT

The eight patterns were elaborated into a new narrative about collaboration. Describing program theories using narratives explains causality in the most concrete way [57]. Clearly identifying the resource mechanisms and multiple contextual conditions interacting at once in the refined MRPT allowed us to identify the key mechanisms promoting outcomes. The research team confirmed the narrative of the refined MRPT after two reflective meetings.

## 7. Ethics

Medical Research Ethics Committee (MREC) approval was received in December 2019 by METC Oost Nederland Number 2019–6022. Next, each policy officer signed an informed consent form for data collection in their case. Finally, each potential participant was given information about the study, and written consent was also obtained per participant per data collection.

All data, including reflections, observations, and timelines, were handled confidentially according to the Dutch Personal Data Protection Act. All data were coded with a consecutive number, replacing any person identification data. Only the researchers (MT and ML) had access to the anonymized transcripts, which were stored on the save directory of the Radboudumc (Cases 2 and 3) and/or Zuyd University of Applied Sciences (Cases 1 and 4). If needed, data were shared between researchers via encrypted e-mail using SURFfile sender.

## 8. Findings

We start this section by explaining contextual factors, mechanisms, and outcomes that led to a refined MRPT for collaboration in developing and sustaining DFIs; see Figure 2. Then, we report on the involvement of people with dementia and their carers in the collaboration for DFIs. We end this section with the narrative of the refined MRPT; see Box 4 and a comparison between the initial theory and refined theory.

## 9. Contextual Aspects

### 9.1. Diversity in Expertise among Organizations and Collaboration Partners

Collaboration partners recruited others from their network who could help to develop and execute DFIs. An important aspect was a diversity in expertise that broadened the perspective on DFIs and necessary resources.


*And I think it is important that in such a working group, you can also compare different experiences. I believe you do that with people from different perspectives and angles, so, different areas. I do think that we are sitting down with the right partners to do that. I think that you should indeed get it done from the collaboration and different expertise, yes.*
(Case 2-T1)

Cases 2–4 were aware that the expertise of people with dementia and carers could be an important addition to others’ expertise. However, participants expressed that current collaboration partners were often occupied getting acquainted and gaining insight into each other’s expertise as they are used to doing. Despite the intention, it left less room to focus on the involvement of people with dementia and carers and their expertise.


*Like we said, we need to bring someone [person with dementia and/or carer] along sometime. Take someone along at least once. But those are not things we have been busy with ... because we are continuously working on our own process.*
(Case 3 T1)

Some collaboration partners were reluctant to invite people with dementia into the collaboration for avoiding contamination during the COVID pandemic.

### 9.2. Shared Insights Regarding the Need for a DFC

In all the cases, there was the aim to become a DFC, expressed earlier by the policy officer during inclusion. Next, collaboration partners supported the need for a DFC and the importance of developing DFIs from their own personal or professional connections.


*Speaker 1: Regardless of one’s background, you have to have something to do with it. Speaker 2: Yes, but I don’t think you would join otherwise. I don’t think we need to spend a lot of thought or attention on emphasizing the importance of a dementia-friendly neighbourhood, because we do agree on that. We all have that vision.*
(Case 2-T1).

Conversely, if the need for a DFC was not recognized by members of the community, a collaboration to develop DFIs would not start, as emerged from the data in Case 1, which withdrew.


*… Plus, I have never received any questions or concerns in my network about dementia- friendly initiatives, or that they very explicitly encountered problems with dementia in X, that it was not properly arranged.*
(Case 1).

What is significant for this contextual aspect was the fact that collaboration partners shared and discussed insights with each other about the needs of people with dementia and carers, leading to ongoing actions. When such insights were missing, proceeding towards developing concrete DFIs was difficult, as the following quotation illustrates.


*Speaker 1: We don’t really talk much about dementia, or about people with dementia. Speaker 2: No, that’s true. SP1: Or about their experiences and such. SP2: I think we are missing … our connection with it. ... We have not actually reached that depth, in my opinion. And, therefore, we keep going in circles.*
(Case 3-T1).

When people with dementia and carers became more involved in the collaboration, collaboration partners felt more convinced about the need for DFC.


*SPV: But I believe in this, that this [collaboration for DFIs with people with dementia and carers involved] could work, yes.*

*SP1: So, what exactly is it [that] you believe in?*

*SPV: That there will be more room for people with dementia and their carers to lead a meaningful and pleasant life in the neighbourhood and to be involved in that process, that is actually what it is about. And what we should always keep in mind, that is the goal it is all about.*
(Case 2-T2).

### 9.3. Clarity about Structure, Roles, Allocated Time, and Budget for Collaboration

A challenging aspect was the clarity with regard to structure, roles, allocated time, and budget. The resources that were implemented provided structure in timely meetings and options to take on a role that suited personal and/or professional interests. However, whether it concerned roles, allocated time, or budget, the most important thing was gaining clarity about that among the collaboration partners by reflecting on this regularly. This provided an overview of what everyone could contribute in the collaboration.


*There must be transparency in the manpower. Who am I at the table, what will they do with what role? Even now, I still have to think: Who has been in that group now? In the beginning, the group was very big, and people have left, and then it just goes away completely. Then I think: What was the role or function of those people in that whole story. You must continue to have a clear picture of what everyone will contribute and what everyone can do.*
(Case 4-T2).

Furthermore, clarity about roles and allocated time was important for transparency about collaboration partners’ contributions to the collaboration, as well as mutual expectations.


*And so, with the few hours that I have, I do try to address issues that exist. I mean, a dementia-friendly neighbourhood is one, but there are many other projects that also take up those few hours. So, I just wanted to say that. I also find that difficult. Yes.*
(Case 2-T2).

Lastly, the need for clarity about roles and allocated time also impacted the perspectives on the involvement of people with dementia and their carers.


*[However], I think that, in particular, we should start talking to each other about if you want to have a mixture of non-dementia and dementia people; it requires a completely different approach. Look, we can facilitate and organize, but where does the support [for people with dementia] come from? Do you want to connect to that? And where do I get the support from? I think that’s the essential question.*
(Case 4-T2).

## 10. Mechanisms

Identified mechanisms were grouped into either “resources” or “responses” to make it transparent that the identified resources (i.e., the recognition of effort and progress, informally distributed leadership, and interdependence) triggered the responses, i.e., belonging, significance, and commitment with regards to collaboration for DFIs. The resources can affect multiple responses. Where data-analyses pointed out a clear connection, this has been described.

### 10.1. Resource: Recognition of Effort and Progress

Recognition of effort meant that all steps in the process to develop and sustain DFIs, regardless of how big or successful they were, were acknowledged by the collaboration partners based on the effort that was put into them. The same went for the recognition of progress; some efforts led to progress, some did not (yet). Even without progress, it was important to take the time to discuss the efforts made and how things could improve from an appreciative point of view.


*If you take the time together to also go over those small steps that you have taken or those small new collaborations, if you share them with each other, then you can easily find new energy or new collaborations, or you can start sparring with each other about taking it one step further.*
(Case 2-T2).

### 10.2. Resource: Informally Distributed Leadership

Informally distributed leadership meant that each collaboration partner could guide or influence parts of the process and decisions during the collaboration for DFI using their own skills or resources, instead of having a hierarchy with a formal designated leader. This meant that each input could be different but was considered equally important for the process of collaboration. Informally distributed leadership was not predetermined but arose during the process of collaboration by acquaintance, using and building a network, having short and informal lines of communication, and appreciation while the facilitator aimed to make himself or herself redundant.


*Yes, and that it consists of different parts in which everyone takes decisive actions. We can indeed organize various initiatives to promote participation [of], as you say, XX. And, as YY urges, something has to be done about that; everyone should know a little bit about dementia or dealing with dementia, and then we go for it. And ZZ then confirms, yes, we can get to that outdoor environment, and so we do that. So, we are all advocates for a part that you need to put in to get the picture complete.*
(Case 4-T2).

### 10.3. Resource: Interdependency

Interdependency during the collaboration for DFI arose through acquaintances, appreciation, network building, and concrete plans for DFIs. Collaboration partners felt that, despite the differences in backgrounds and abilities, there was a reciprocity and equity in need of each other, and that everyone made an important contribution to the realization of success.


*Speaker A: That is also about insight into relationships … if you would say very bluntly: Okay, B is from the municipality. You can see that. [However], how does that compare? I just say it like it is. [However], then there is also: B also has the same interests. The same goal. A nice person. So, it’s about: I need B. Speaker B: Yes, and I need S, and I need you all. Speaker A: That’s so important, we always need each other.*
(Case 3-T2).

Here, the position of people with dementia and carers was characterized by ambiguity. On the one hand, people with dementia and their carers were acknowledged for having a unique perspective that was needed; on the other hand, collaboration partners felt that they were informed well enough by their own experiences and consultation with people with dementia and their carers.


*[In addition], I think it is important that, in such a working group, you can also compare different experiences. I can imagine that they [people with dementia and their carers] may have a different image or have different experiences. What are we talking about then? Well, I believe you do that with people from different perspectives and angles, so, different areas. I do think we could reinforce that by broadening that perspective even more by involving people with dementia and carers, but we’ve talked about that before. [However], I still think that we are sitting down with the right partners to do that now*
(Case 2-T1).

### 10.4. Response: Belonging

A very powerful response was the sense of belonging in the collaboration for DFI. Collaboration partners felt connected with each other, the content of the DFI, or the working process due to acquaintance, short and informal lines of communication, appreciation, and the detailed plans for DFIs. It became a powerful response to either stay in or drop out of the collaboration.


*As volunteers, we may also be “hands-on” people, meaning you want to get to a result quickly and have that result quickly on the table. [In addition], that is also one of the reasons that I joined the DFI of Movement, because I thought: now I have something concrete, let’s get started.*
(Case 2-T1).

Conversely, a caregiver of a person with dementia who first joined the collaboration withdrew because she did not feel comfortable in the working process.


*On my initiative, I was presented as a contact person for the carers’ café. At first, I thought it was a good idea because I know a lot of people in X. However, this unintentionally created the impression that I would have a leading role in the carers’ cafe. That made me uncomfortable. ... I feel out of place among you all, though you are such warm and supportive people.*
(Case 3-T1).

### 10.5. Response: Commitment

Commitment was expressed as the intention to play an active role in the collaboration for the DFI, e.g., to take responsibilities in the development and/or the execution of the DFI. Commitment was action-oriented and arose by making detailed plans for the DFI, having short and informal lines of communication, appreciation, and building a network. It could be connected to the purpose of the collaboration, namely developing DFIs, or to the other collaboration partners.


*Speaker 1: That’s where my heart lies, doing this. So, this is double for me. This is actually my job, but I also find it really fun to do. It’s actually half work for me, half my own ambition, drive, the desire to improve things for people with dementia and carers a bit in X. Speaker 2: What made it valuable to you to make time for it? Speaker 1: Well, exactly what also has been stated, you just see that here are people who want to. Those who are ready and willing to help. [In addition], I want to help them also.*
(Case 3-T2).

Commitment was sometimes expressed in a critical way, through concerns about the commitment of other stakeholders in the process.


*When we started, A was naturally involved, and he once showed me in the policy documents that dementia and dementia-friendliness was a topic on the municipality’s agenda. I would be curious to find out how B feels about it now.*
(Case 4-T2).

### 10.6. Response: Significance

The sense of significance is the feeling that one’s presence, efforts, or actions would matter or be acknowledged as adding value to achieve the goal of the DFI. When plans for DFIs were concrete or communication lines were short, this was very helpful. Conversely, when these ingredients were not present, people did not feel that their efforts and actions had any impact or were recognized. Feeling a sense of significance strongly influenced whether collaboration partners remained in or withdrew from the collaboration.


*In the beginning, I personally had the idea: well, let’s get it done quickly. The longer it lasted, the more disappointed I actually became. [However], it depends on what it is. Sometimes, the process [towards reaching DFIs] may be unclear… but I really would love to see or hear that our work is not useless but noticed.*
 (Case 3-T1).

When people had an idea of what the DFI could look like, they also had an idea of what their contribution could look like.


*The degree of concreteness of whether someone can visualize this in their own environment so to speak. What does it mean for me? How can I be of use? Yes, then perhaps people are more inclined to say: Yes, I will participate in that.*
(M-T1).

## 11. Outcomes

Within the timescale of this evaluation, the outcomes of collaboration for DFIs perceived by the collaboration partners were on the personal level: activation, fun, and new ideas. The outcomes are also closely related to each other, each representing a specification of positive energy during the collaboration for DFIs.

### 11.1. Activation

Activation refers to taking action for a DFI, either by executing DFIs or by creating conditions that enable the implementation of DFIs in the community. It is closely related to the response of feeling significant, which was a strong motivator for concrete activation.


*At that working group meeting, we talked about this community [CoP] and our ideas as well. It makes everyone more open to our DFI. Employees of the daycare thought: Oh, yes, this is something we can do there. Then I think: We can get people to participate in our activities [DFIs]. Then I think, yes, set a date and let’s have a go.*
(Case 3-T1).

### 11.2. New Ideas

Getting new ideas generated a positive vibe and was a two-sided outcome. It was an outcome for both the collaboration partners and for the purpose of the collaboration, namely the DFIs.


*[In addition], I really liked the added value of the students. They were really, very actively involved. [Plus], they came with new ideas, and I think without those ideas, the start-up would have been much more difficult. I am convinced of that. [However], also … everyone had ideas from different perspectives. I think that also makes it a lot of fun.*
(Case 2-T2).

### 11.3. Fun

Fun represented an outcome of personal interest of the individual collaboration partner rather than an outcome of interest of the group of people with dementia and carers.


*Speaker 1: The meetings for the informal care café are always very spontaneous, very pleasant, very nice. So, that was no overload at all. … Speaker 2: I recognize that. I liked the involvement; I remember very much that we had a meeting at your house. I don’t know, that was very informal or something, B. who took a leading role, and M. who is always present. Yes, I don’t know, that gave a lot of positive energy.*
(Case 3-T2).


*When is the collaboration a success? When you get along well. [In addition], when you go home, you have a good feeling. You say: Well, it was nice, it was fun.*
(Case 2-T2).

## 12. Report on the Involvement of People with Dementia and Their Carers in the Collaboration for DFIs

Based on the minutes, reflections, and observations, and the data from the focus groups, the involvement of people with dementia and their carers in the collaboration for DFIs is illuminated, using the typology of citizen participation of Arnstein [48]. Table 3 shows the levels of participation of people with dementia and their carers during the collaboration for DFIs using the typology of Arnstein.

## 13. Refined MRPT of Collaboration for Development and Sustainment of DFIs

The following refined middle-range program theory was developed from the findings above. It is specified in Box 4.

Box 4Narrative of refined middle-range program theory about collaboration.Refined middle-range program theory: Collaboration for developing and sustaining DFIs For collaboration in developing and sustaining DFIs, professionals and volunteers with diversity in expertise and organizational background come together during
different ways of collaboration, preferably complemented by people with dementia and their carers. They all have a connection with dementia, share
their insights about the needs of people with dementia and their carers with each other, and agree on the importance of a DFC. As a result, involved
professionals and volunteers understand the urgency of the collaboration and conditions necessary for it, including the involvement of people with
dementia and their carers during the collaboration for DFI. Other contextual features are clarity about structure, roles, and allocated time and budget during
the collaboration for DFI. During structured meetings, allocated and possible timeframes, individual perspective regarding roles, and budget are discussed.
Other important actions in these meetings are getting acquainted with each other and building a network, along with people with dementia and their
carers, using short and informal lines of communication, appreciating each other for the efforts made and making detailed plans for concrete DFIs.
Hence, mechanism resources such as recognition of efforts and progress, informally distributed leadership and interdependency arise. These contextual
features and mechanism resources lead to mechanism responses of belonging, commitment, and significance. Outcomes of collaboration for DFIs were activation, gaining new ideas, and fun.

## 14. Similarities and Differences between the Initial and the Refined MRPT for the Collaboration for Development and Sustainment of DFIs

Most of the components of the initial MRPT were confirmed in this study. The contextual factors identified in the new MRPT, i.e., diversity in expertise and organizational background, preferred complemented by people with dementia and their carers, a personal or professional connection with dementia, clarity about structure, roles, time, and budget are also visible in the initial MRPT. Mechanisms in the refined MRPT, such as informally distributed leadership resonate with the initial MRPT, in which actions such as taking a role that best suits personal and professional experiences, sharing, and joint decision-making, are incorporated. The refined MRPT also confirms commitment, activation, and fun as important characteristics in the collaboration.

The refined MRPT differs from the initial MRPT, as it elaborated more on the involvement of people with dementia and their carers during the collaboration, e.g., the need to involve people with dementia and the making acquaintance and network-building with people with dementia and their carers. Furthermore, the refined MRPT elaborates more on actions during the collaboration, such as discussing timeframes, personal interpretation of roles, and budget, in addition to getting acquainted with each other, building a network, using short and informal lines of communication, and appreciating each other for the efforts made. To conclude, the refined MRPT addresses mechanisms such as interdependence, commitment, belonging, and significance, as well as outcomes such as gaining new ideas.

## 15. Discussion

Collaboration for DFIs to build DFC and, thereby, address health disparities is found to be complex and context specific [6,13,16,21,28]. By using a realist evaluation design, we gained a more in-depth understanding of how collaboration works to develop and sustain DFIs in a local public context. By specifying the causal relationships between context, mechanisms, and outcomes, and the involvement of people with dementia and carers therein, this is, to our knowledge, the first study to take into account this kind of complexity to understand how collaboration for DFIs may work. The results of this study provide detailed information about the influence of local contexts and subsequent mechanisms and outcomes on the collaboration for DFIs and the involvement of people with dementia and their carers in collaboration for DFIs.

Mechanisms in our refined MRPT, such as commitment, interdependency, and (informally) distributed leadership were consistent with previous literature about collaboration [6,10,18,21,26,27,58]. However, rarely mentioned in literature are the mechanisms about recognition of effort and progress, significance, and belonging. These mechanisms resonate with feeling useful and feeling collectively powerful.

In contrast to other literature that mostly describes what needs to be undertaken for developing DFIs [6,13,16], our study provides more insights into the mechanisms that create change during the process of collaboration. The refined MRPT showed a rearrangement of actions, mechanisms, and outcomes compared to the initial MRPT. For example, mechanism resources, such as feeling connected with each other in the initial MRPT, were further elaborated as actions, such as building a network and appreciating each other, and mechanism responses, such as a feeling of belonging in the refined MRPT. In addition, actions, such as joint decision-making in the initial MRPT, were elaborated as resources, such as informally distributed leadership, and mechanism responses, such as commitment in the refined MRPT. These clarifications are important to understand the dynamics of collaboration [58] with a differentiation between actions undertaken and mechanisms that underpin change during collaboration [33,36].

Further research is needed to elaborate on the involvement of people with dementia and their carers in collaboration for DFI. Our study shows that their involvement in the collaboration was very limited despite stressing its importance by the facilitators. As such, people with dementia and their carers had limited power to influence or change the collaboration for DFI. According to Arnstein [48], their involvement can be characterized as tokenism during informing, consultation, and placation. Some degree of citizen power was visible within Cases 2 and 3 during partnering and delegated power, by which people with dementia and carers could influence the collaboration for DFI. The contextual factors and mechanisms reveal how the involvement and expertise of people with dementia and carers seem to be overshadowed by a professional hierarchy, hunches and work routines. The lack of involvement of people with dementia and their carers during the collaboration for DFIs represents exclusion and reflects a social injustice that perpetuates health inequalities [8].

Our findings regarding the limited involvement of people with dementia and their carers align with other studies about the involvement of people with dementia and their carers during the collaboration for DFC or DFIs [17,43,59,60,61,62]. Our study also aligns with other studies by the way of organizing the collaboration; e.g., how the structure and inclusion of collaboration is much influenced by the initiators and consortia. Therefore, the inclusion of collaboration partners starts mostly from organizations, as in our study.

The typology of Arnstein is very useful to discuss the power of people with dementia and their carers and address ways to improve power in collaboration. However, our results also showed that informally distributed leadership made a carer feel uncomfortable. Although the intention to share power with people with dementia and their carers cannot be disputed, it raises the question about preferences of people with dementia and their carers on how to be involved in the collaboration for DFIs. Next, the typology of Arnstein has been criticized for the absence of context sensitivity, e.g., how the typology might be used as a collective progress between all stakeholders involved [63,64]. Studies from a research context showed that the involvement of people with dementia and their carers supported mutual learning; enhanced impact on researchers’ behaviours, emotions, and values had a positive influence on the quality of the research and made it highly context specific [65,66,67]. The studies show how the involvement of people with dementia and carers was organized around mutual benefits rather than power dynamics. Our results suggest a learning process between the first and second focus groups regarding the need for the perspective of people with dementia and their carers in the collaboration. This was visible in the results on contextual aspects, namely diversity in expertise among organizations and collaboration partners, shared insights regarding the need for a DFC among collaboration partners, and clarity about structure, roles, and allocated time and budget for collaboration. Lastly, the current collaboration partners found clarity about time, budget, and roles challenging. This also applies for people with dementia and their carers in how they want to manage their lives and avoid overburden and distress [68,69]. This could influence their participation level in the collaboration for DFI. Subsequently, our outcomes, such as activation, getting new ideas, and experiencing fun, refer to the positive energy that is associated with collaboration, which are also important outcomes for people with dementia and their caregivers to feel included [20]. For that reason, it is also important to involve them in the collaboration for DFIs.

## 16. Strengths and Limitations

Based on the hallmark criteria for a realist evaluation [30,38,70], our evaluation has an explanatory focus by (a) Reformulating and refining the MRPT; (b) Investigating linked configurations of context(s), mechanism(s), and outcome(s); (c) Using multiple methods of data collection; (d) Including stakeholder involvement by engaging an advisory panel; and (e) Aiming for data cumulation rather than replication by the use of joint learning and sharing interim findings in the CoPs [30,36,70,71]. Furthermore, the inclusion of exit interviews in data collection, reflections, and observations in focus groups, as well as rival statements during theory testing, added a critical perspective on collaboration [53]. Testing theories during the COVID pandemic brought new context elements, such as restrictions to come together, that impacted on the mechanisms. We took it as an opportunity to expand and refine our thinking about the role of context and mechanisms [72,73]. By disaggregating the resource mechanisms from the reasoning mechanisms, key resources that triggered reasoning to enable specific outcomes were identified. These resource mechanisms have important practical application for facilitators and other implementers in terms of establishing positive conditions for change for all collaboration partners involved. The reasoning mechanisms identified help to deepen our understanding of how and why resources introduced under specific contextual conditions are likely to bring about outcomes and for whom [34,39].

With regard to our aim to deepen insights into local contexts, our study did not prescribe how people with dementia and their carers should be involved. Reflecting on our approach, we aimed to gain mutual benefits in the collaboration. The research team considered this as a part of the local and plural aspects of the contexts and a way to include the diversity of the population of people with dementia. Instead of a pre-scripted method, the facilitator encouraged collaboration partners to act and reflect on the involvement of people with dementia and their carers. This led to valuable insights into the contextual aspects and mechanism about involving people with dementia and their carers. It also underlined the transformative leap that is required in dementia services and support from the legacy of service-led working cultures into a new provision of dementia services and support that has its origin in the perspectives and frame of reference of people with dementia and their carers.

Special attention should be paid to the fact that data was collected during the COVID-19 pandemic, a period in which the urgency for DFIs increased because of exacerbated difficulties for people with dementia, such as the deprivation of activities and loneliness [74,75], while, simultaneously, policy was directed only towards acute health delivery [76]. This made collaboration partners sometimes uncomfortable due to competing areas of attention. The impact of COVID-19 also made collaboration partners reluctant to invite people with dementia and their carers to join the collaboration. Additionally, a few collaboration partners considered the transition to online CoP meetings beneficial because it saved travel time and was, therefore, more convenient to fit into busy agendas. However, most collaboration partners questioned whether it deepened the connection between collaboration partners and raised concerns regarding the greater risk for withdrawal. Next, in all cases, collaboration partners discussed if the development of technology-based DFIs could be an appropriate alternative to existing plans. However, all cases regarded their resources in terms of expertise, funds, and time as being too critical. Instead, they collaborated towards DFIs that were possible within the COVID-19 pandemic restrictions, such as walking routes. All these experiences add significant strength to the findings about collaboration, as we identified under both positive and negative contextual conditions how the mechanisms worked.

Some challenges were experienced in terms of data collection. Only four cases and two cycles of data collection were possible; more would have allowed further testing of theories [30,37].

Our study had dropouts of professionals, volunteers, entrepreneurs, and carers. Volunteers, entrepreneurs, and carers were not recovered. The mechanism interdependency bridged differences between professionals and volunteers, as diversification and social closure can affect their collaboration [77]. However, it is important to gain more understanding how interdependency works and for whom. Since, in our study, collaborations were in the early stage and DFI was a new topic in the community, these are important findings for understanding how to keep people connected and included in the initial phase of the collaboration. For insight into collaborations at later stages, a long-term follow-up study is needed.

## 17. Future Research

This study built on the results of four cases, which provided the context of municipalities with an ambition to become dementia- friendly. As there are many other municipalities with the same ambition that are collaborating for DFIs, further research, including additional contextual differences is needed. Further research should focus on the personal and professional background of all stakeholders in the collaboration, including people with dementia and their carers, regarding diversification and social closure. It will deepen the understanding of what diversity brings about in the collaboration and how, other than expertise and resources, and for whom collaboration is successful. Furthermore, the involvement of people with dementia and their carers in collaboration needs attention, especially how they favour their involvement, mutual benefits, and power dynamics. For future research into collaboration with DFIs, it is crucial to understand how mechanisms on collaboration for DFIs can be activated with people with dementia and their carers, as these mechanisms also represent the ambition of a DFC. Especially belonging is the goal of an inclusive dementia-friendly community and turned out to be an important mechanism in our study. It is an important finding for mostly professionals and volunteers. Therefore, it is crucial to understand how this works for people with dementia and their carers [8]. Future studies will be needed to further test and refine our MRPT in other cases and contexts. Additionally, research is needed to investigate perspectives of all stakeholders, including people with dementia and their carers, on the involvement of people with dementia and their carers during collaboration.

## 18. Conclusions

The collaboration for the development and sustainment of DFIs is found to be complex and context specific. Our realist study is characterized by testing an initial theory about collaboration on developing and sustaining DFIs by using data from a participatory action research of four municipalities. The collaboration for DFIs was largely influenced by contextual aspects such as diversity, sharing insights, and clarity next to mechanisms such as recognition of efforts and progress, informal distributed leadership, interdependency, belonging, significance, and commitment. The outcomes were activation, getting new ideas, and fun. Stakeholders’ routines and perspectives influenced how people with dementia and caregivers were (not) involved. The insights from this study can help municipalities with an ambition to become dementia-friendly to start up a collaboration for DFIs. Our findings stress the need for more insights into the collaboration for DFI with people with dementia and carers. Such insights will help achieve the ultimate aim to create evidence-based inclusive, dementia-friendly communities and strategies.

## Figures and Tables

**Figure 1 ijerph-20-04006-f001:**
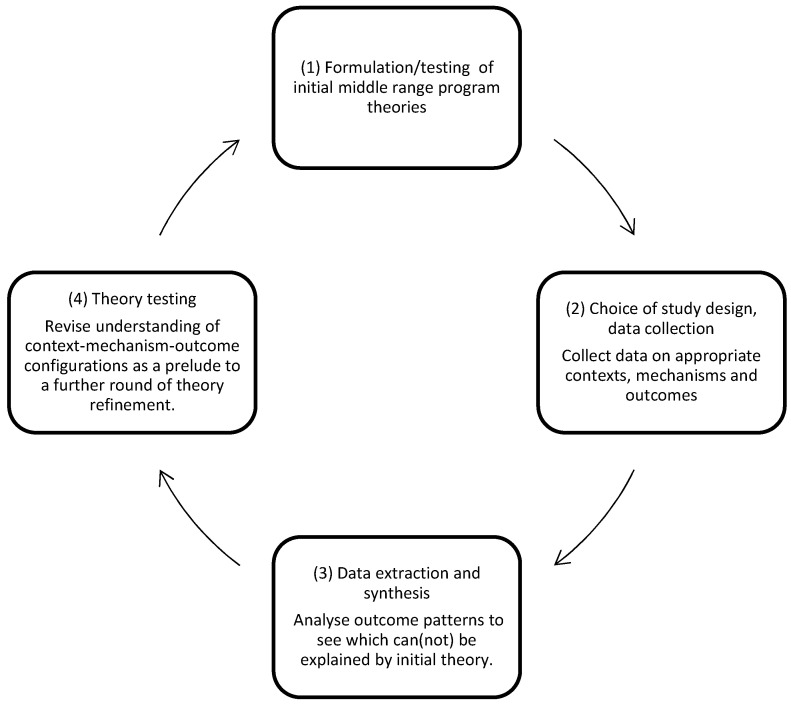
Realist cycle; adapted from Pawson [30] and van Belle et al. [37].

**Figure 2 ijerph-20-04006-f002:**
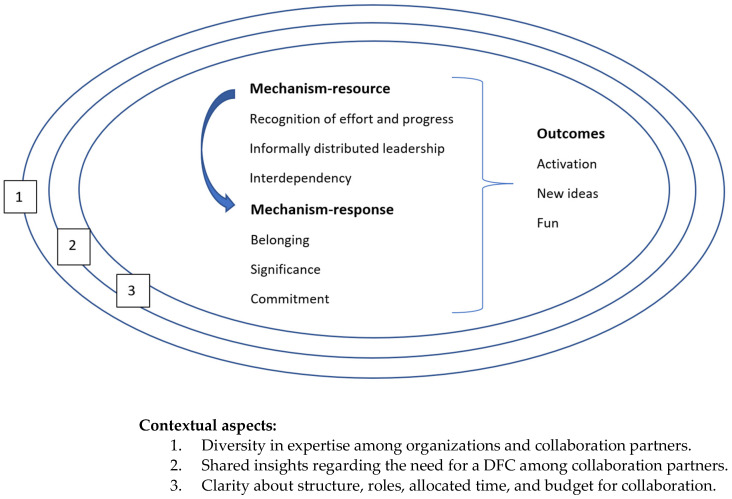
Refined MRPT for collaboration in developing and sustaining DFI.

**Table 1 ijerph-20-04006-t001:** Overview of collaboration partners for DFIs in the four cases.

	CS 1 (Rural)		CS 2 (Urban)		CS 3 (Rural)		CS 4 (Urban)	
	Included	Withdrawal	Included	Withdrawal	Included	Withdrawal	Included	Withdrawal
Healthcare and social professionals	3 *	3 *	5 ***	0	3 ***	0	10 ***	2 *** (replaced)
Volunteers/community members	0	0	3 *** 1 ****	1 * (not replaced)	7 ***	4 ** (not replaced)	7 ***	0
Entrepreneurs	0	0	1 ****	0	1 *	1 * (not replaced)	1 ****	0
Policy officers	1 *	1 *	1 *	0	1 *	0	1 *	1 * (replaced)
Students occupational therapy	0	0	4 ****	0	4 ****	0	0	0
Carers of people with dementia	0	0	1 * 2 ****	0	1 **	1 ** (not replaced)	0	0
People with dementia	0	0	1 ****	0	0	0	0	0
**Total collaboration partners**	**0**		**14**		**7**		**17**	

CS = case. Superscript * member CoP, ** member COP and working group, *** member COP and working group and collaboration partner during implementing DFIs **** collaboration partner during implementing DFIs.

**Table 2 ijerph-20-04006-t002:** Presentation of the DFIs that were developed or sustained in each case, and the participants during data collection.

	CS 2	CS 3	CS 4
**DFIs**	1: Walking route 2: Stories of people with dementia in a district paper 3: Connecting groups in community 4: Training dementia friendliness	1: Walking route 2: Carers’ cafe	1: Memory information desk 2: Music group 3: Training dementia friendliness
**Number of participants per focus group July 2021**	**3**	**4**	**4**
Healthcare and social-work professionals	1 **	1 **	1 **
Volunteers/community members	1 *** 1 ***	2 **	3 **
Entrepreneurs	*0*	*1 ** ***	*0*
Policy officer	*0*	*0*	*0*
Carers of people with dementia	*0*	*0*	*0*
People with dementia	*0*	*0*	*0*
**Number of participants in focus group December 2021**	**9**	**8**	**5**
Healthcare and social-work professionals	1 * 3 ** 1 ****	1 * 1 * 2 ***	0
Volunteers/community members	2 ***	3 ***	1 ** 3 ***
Entrepreneurs	0	0	0
Policy officer	1 *	1 *	1 *
Carers of people with dementia	1 **	0	0
People with dementia	0	0	0

CS 2, 3, 4 = Case 2, 3, 4. Superscript * member CoP, ** member COP and working group, *** member COP and working group and collaboration partner during implementing DFIs **** collaboration partner during implementing DFIs.

**Table 3 ijerph-20-04006-t003:** Levels of participation of people with dementia and their carers during the collaboration for DFIs using the typology of Arnstein.

Level of Participation according to Arnstein [48]	Activities during the Collaboration for DFIs
Citizen control People with dementia and carers have the idea and set up the DFI.	n.a. (not applicable)
Delegated power Goal created by CoP but responsibilities and resources given to people with dementia and carers.	People with dementia and carer shared their stories for the district paper (Case 2). Carer was a co producer of the carers café (Case 3).
Partnership Stakeholders have direct involvement in decision-making.	Carer was a member of the CoP (Case 2).
Placation People with dementia and carers shape ideas but professionals and volunteers decide.	Person with dementia and carer visited the CoP and provide suggestions for DFI (Case 2).
Consultation Views of people with dementia and carers are sought but professionals and volunteers decide.	People with dementia and carers were interviewed by professionals about their needs (Cases 2 and 4).
Informing People with dementia and carers are informed but have no opportunity to contribute.	People with dementia were informed about the DFIs (Cases 2, 3 and 4).
Therapy Assumption that people with dementia and carers are recipients.	n.a. (not applicable)
Manipulation People with dementia and carers are denied of power.	n.a. (not applicable)

## Data Availability

Data is contained within the Appendix A. The data presented in this study are available in Appendix A: Eight patterns cross cases.

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
