# Peer review of "Collaboration for Developing and Sustaining Community Dementia-Friendly Initiatives: A Realist Evaluation"

_ijerph, 2023, doi:10.3390/ijerph20054006_

Round 1

Reviewer 2 Report

Many thanks for the opportunity to review this paper. It offers a useful and detailed account of efforts to evaluate the process of planning and preparation for dementia friendly community development. The paper will be of interest to the readers of this journal and will make a contribution to the wider emerging debate on the now international programme of dementia friendly development. I have three main issues I’d like to highlight for consideration when it comes to your revisions of this paper.

1.      You’ve outlined quite a detailed and complex narrative involving the setting up and subsequent evaluation of the so-called Communities of Practice (CoPs) as well as offering some insight into their workings and operation. While the level of detail is very welcome (you note yourselves that much existing literature tends to focus on generalised principles or activities) it would be worthwhile to prioritise clarity in your revisions of the paper. From a reader’s perspective there are passages to this paper that are quite hard work. In part this is due to the range of different ideas, concepts and theoretical perspectives that your commentary draws upon. However, I think the use of initialised/acronym terms, and (over?) emphasis on description rather than explanation is also a factor. Particularly in the section where you are presenting your methodology, I found myself asking whether the diagrams and boxes were helping to simplify the narrative or adding yet another level of abstraction. My main advice in this respect would be to put a little more emphasis on concrete examples to help anchor the ideas being shared and to ensure that you are explaining meanings, decisions, and interpretations along the way rather than just describing the process of what you did. I also wonder if some of the detail can be weeded out a little – there’s a lot of fairly complex information to absorb. Greater clarity would also be helpful early on about what is being evaluated here. My initial assumption was that the focus would be on dementia friendly community development but as the paper progresses the focus seems to be much more narrowly upon the series of meetings held by each CoP. Thus, later on when you talk of outcomes, these aren’t the outcomes of the work being done in each local area but outcomes of the meetings for those involved. This raised a question for me as to the meaningfulness or significance of meeting outcomes when they’re not tied to actual change within the communities they serve or to any evidence of concrete outcomes for those communities (it’s great that these practitioners are having fun but so what if that doesn’t lead to change for people with dementia?). A further question is to what extent this study is actually then about DFCs, because overall there is very little evidence or insight into anything that might be considered dementia friendly anywhere in this paper, other than the expressed commitments of the participants whose own claims are somewhat undermined by their practices. As I’ve detailed below this provides grounds for pushing back a little at your current interpretation especially in respect to your suggestion that the study has demonstrated the efficacy of collaboration in developing and sustaining a DFI. Indeed, you’ve made claims that the research demonstrates the sustainability of the initiatives but how has this been gauged? Is this an unsupported claim given the early stage of the project and time-limited nature of the evaluation?

2.      A second consideration concerns questions of coherence for the paper. My overall impression is of a disconnect between the opening and closing section compared to the section where the findings are presented. There are also a number of points picked up early on that aren’t further developed or returned to as the paper progresses. For instance, you open with attention to questions of health disparities but do not return to this in your subsequent discussion so it’s unclear what this study contributes to this topic. If your suggestion is that DFCs offer an approach to address such disparities then this line of thinking needs further fleshing out (it certainly hasn't been demonstrated by your findings). You also mention an urban and rural split of case studies but there’s no subsequent discussion or analysis of this, despite your claim that context includes geographical location effects. One of the main questions I have about the paper is that while you put great emphasis on involvement of people with dementia as a foundational criteria for DFCs at the outset of the paper and return to this in your discussion, this stance isn’t maintained in the presentation of findings. There is an absence of any real analysis of the exclusionary nature of the CoPs. This is why the paper feels a little fragmented and I would suggest that you have perhaps underplayed the implications of the exclusion of people with dementia in reaching your judgements as to the relative success of the CoPs. As a reader my questions were 1) why was no provision made for the support needs of people with dementia in the setting up of the CoPs? 2) why didn’t the researchers more actively question and probe the participants in respect to their exclusionary views and practices toward people with dementia 3) why didn’t the researchers correct for the exclusion of people with dementia by interviewing people living within the areas where the DFI’s were taking place? And, 4) why isn’t this significant omission the subject of further analysis in the paper itself?

I also noted early on your emphasis on the importance of incorporating local context as another core aspect of the DFC/I process (which I applaud). But I struggled to see how this local context was made manifest in the CoPs or indeed by the research. There is a question here of how ‘context’, as defined in your opening to the paper, is then operationalised by the fieldwork. Obviously if context is seen as an objective, singular phenomenon then the knowledge of the CoP participants may perhaps suffice but if we understand context (or the localities in which the DFIs are being developed) as contested or plural then clearly the absence of the lived experience of people with dementia is a major omission.

3.      Consequently, the third point I wanted to raise was in respect to your interpretation of the case studies and their achievements. My take-away from this paper is that it offers a narrative of a project that set out to create dementia-friendly conditions at a local level but failed in doing so, to the extent that the CoPs were anything but dementia friendly (we’re not given any evidence of the actual community development work to judge beyond the CoPs). In this respect it echoes findings from other studies (e.g. Buckner et al) that point to the legacy of a paternalistic and exclusionary service-led culture but I’m not sure that’s the reading this paper is offering. For instance you talk of belonging as being an outcome of the CoPs whilst outlining the total absence of people with dementia and sharing a quote from a carer who dropped out of the CoP on the grounds that they were made to feel they did not fit in. There is a key point in your findings that underlines quite how problematic the working culture and practices of the case studies are when participants claimed to have all the knowledge they required on behalf of people with dementia so as not to involve them directly. They throw doubt on the contribution that people with dementia would make and showed a refusal to adapt their working methods in order to enable people with dementia to participate. The only reference to involvement signals a tokenistic effort of ‘bringing someone along sometime’. There seems to be a real absence of understanding about what co-productive and participatory working involves within these CoPs. Despite this the members (and the authors it would seem) are making claims to this being a dementia-friendly initiative. My concern here is that you are minimising the implications of this exclusion rather than treating it as a prerequisite of DFC/I practice.

Both the CoP participants and the authors have elided people with dementia and carers as if they are an homogenous group and have shown little consideration for the distinct needs and perspectives of each group let alone for the diversity that characterises the dementia population itself. You state that there has been limited involvement of people with dementia and carers in the CoPs when actually people with dementia are entirely absent. Perhaps then, there is an argument to reconsider the way in which you present and discuss these case studies? Indeed, the quote from the carer (I feel out of place among you all) not only seems to capture the working culture that has evolved in these groups but touches on a bigger issue of how the expertise by experience of people with dementia and carers is overshadowed by a professional hierarchy that rules out certain ways of knowing and forms of knowledge. I wonder if this particular comment might make a good title for the paper too? My impression is that your study underlines the scale of the translational and transformative leap that is required in dementia services and support from the legacy of service-led working cultures derived from institutional care settings that are proving entirely outmoded for contemporary provision of dementia services and support. It’s difficult to imagine a practitioner in the field of disabilities claiming there was no need to involve disabled people in a project on inclusion and accessibility and yet, as you demonstrate, that mindset remains in dementia care. Obviously it’s up to you how far you wish to anticipate these criticisms in the revision of your paper but I think (international) readers will justifiably raise questions along these lines. Finally, you may wish to reconsider the gendered language used in some places, i.e. spokesmen, manpower.

Reviewer 3 Report

1. Academic writing: please check all the abbreviations and full names and use them correctly, such as Dementia-friendly communities (DFCs) in line 61.

2. What is CMOC represents should be noted in Figure 1.

3. Box1: what is the context, mechanisms and outcomes in current study?

4. The theory in BOX 2 is suggested to be described in diagram to facilitate reading and understanding.

5. BOX4 is unnecessary to explain these definitions.

6. It is suggested that the BOX 2 and BOX 5 are presented together in order to make a clear comparison.

7. To improve the reporting quality, Consolidated criteria for reporting qualitative research (COREQ) is suggested to follow.

Author Response

Feedback reviewer 3.

  1. Academic writing: please check all the abbreviations and full names and use them correctly, such as Dementia-friendly communities (DFCs) in line 61.

Response 1: thank you for pointing out the grammar and accuracy of academic writing. We have checked the manuscript for abbreviated and made the necessary corrections.

  1. What is CMOC represents should be noted in Figure 1.

Response 2: we have removed the abbreviation and replaced this with the full  description See page 7, Figure 1 on line 164-165 ‘context-mechanism-outcome configurations;  .

  1. Box1: what is the context, mechanisms and outcomes in current study?

Response 3: Thank you for pointing out that this may be unclear for the readers. In box 1; context, mechanisms and outcomes are defined and we provided examples at context, mechanisms and outcomes. See page 6, line 156, box 1:

Context includes the pre-existing organisational structures including the nature and scope of pre-existing (in)formal networks; the cultural norms and history of the community; such as traditions and habits and former relevant experiences, such as experience with dementia-friendly initiatives.

Mechanisms are ‘triggered’ when (program) resources (e.g., information about collaboration, expertise in dementia) interact with specific features of the context (individual, interpersonal, organizational, or institutional).

Outcomes are either intended or unexpected intervention outcomes, i.e., the result of how people react to the mechanisms e.g. taking initiative during collaboration

  1. The theory in BOX 2 is suggested to be described in diagram to facilitate reading and understanding.

Response 4: We present our initial theory by a narrative according to Funnel and colleagues  [1] The narrative of our initial theory describes key parts and the essential features of both actions and change [1]. By describing the theory as a sequence of events and with details if needed, it becomes more accessible and understandable for someone who was not informed about the concept of collaboration for DFI [1]. As such, it aligns with the RAMESES II reporting standards which state that a program theory should explicate how a program is expected to cause its intended outcomes, preferably supported by a diagram or figure [2] To facilitate reading and understanding we added the figure that was used in our previous study. The figure and corresponding description are described in our second study of Mentality that has been accepted for publication by BMC Public Health.  

  1. BOX4 is unnecessary to explain these definitions.

Response 5: We understand that the definitions might be familiar for your readers. However, from former publications and presentations, we noticed that not all researchers feel familiar (enough) with the realist approach and its definitions. Therefore, we suggest that that box 4 will remain in the manuscript to improve transparency of the method and to avoid long definitions in the text that will affect readability.

  1. It is suggested that the BOX 2 and BOX 5 are presented together in order to make a clear comparison.

Response 6: Thank you for pointing out the importance of comparison. Box 2 represents the initial theory at the start of the realist evaluation cycle and box 5 describes the refined theory at the end of the realist cycle. To support the overview of the realist cycle and the results, we suggest that these boxes should not be presented next to each other, that is together at the start of the realist cycle or together at the end of the realist cycle. We understand the importance of comparison and therefore we described the comparison in the section ‘Similarities and differences between the initial and the refined MRPT for the collaboration for development and sustainment of DFIs’ on page 30-31, line 725-744.

  1. To improve the reporting quality, Consolidated criteria for reporting qualitative research (COREQ) is suggested to follow.

Response 7: We added the COREQ checklist in the submission. See Appendix 1 Thijssen Reporting standards - COREQ. In our first submission we included the reporting standards of RAMESES for reporting on the criteria of realist evaluations [2].

  1. Funnell, S.C. and P.J. Rogers, Purposeful program theory: Effective use of theories of change and logic models. Vol. 31. 2011: John Wiley & Sons.
  2. Wong, G., et al., Quality and reporting standards, resources, training materials and information for realist evaluation: the RAMESES II project. Health Services and Delivery Research, 2017.
